# Evolution of alternative biosynthetic pathways for vitamin C following plastid acquisition in photosynthetic eukaryotes

Glen Wheeler[1]*, Takahiro Ishikawa[2], Varissa Pornsaksit[3], Nicholas Smirnoff[3]*

[1]Marine Biological Association, Plymouth, United Kingdom; [2]Department of Life Science and Biotechnology, Shimane University, Matsue, Japan; [3]Biosciences, College of Life and Environmental Sciences, University of Exeter, Exeter, United Kingdom

**Abstract** Ascorbic acid (vitamin C) is an enzyme co-factor in eukaryotes that also plays a critical role in protecting photosynthetic eukaryotes against damaging reactive oxygen species derived from the chloroplast. Many animal lineages, including primates, have become ascorbate auxotrophs due to the loss of the terminal enzyme in their biosynthetic pathway, L-gulonolactone oxidase (GULO). The alternative pathways found in land plants and *Euglena* use a different terminal enzyme, L-galactonolactone dehydrogenase (GLDH). The evolutionary processes leading to these differing pathways and their contribution to the cellular roles of ascorbate remain unclear. Here we present molecular and biochemical evidence demonstrating that GULO was functionally replaced with GLDH in photosynthetic eukaryote lineages following plastid acquisition. GULO has therefore been lost repeatedly throughout eukaryote evolution. The formation of the alternative biosynthetic pathways in photosynthetic eukaryotes uncoupled ascorbate synthesis from hydrogen peroxide production and likely contributed to the rise of ascorbate as a major photoprotective antioxidant.

*For correspondence: glw@mba.ac.uk (GW); n.smirnoff@exeter.ac.uk (NS)

**Competing interests:** The authors declare that no competing interests exist.

## Introduction

Ascorbate (vitamin C) plays an essential role in eukaryotes as an enzyme co-factor in hydroxylation reactions, contributing to diverse processes such as the synthesis of collagen and the demethylation of histones and nucleic acids (*Mandl et al., 2009*; *Blaschke et al., 2013*). Ascorbate also plays an antioxidant role in eukaryotes to help protect against reactive oxygen species (ROS) derived from metabolic activity. The majority of hydrogen peroxide ($H_2O_2$) generated in some organelles is likely reduced by other antioxidant systems, such as the peroxiredoxins and glutathione peroxidases in the mitochondria, and catalases in the peroxisome (*Mhamdi et al., 2012*; *Sies, 2014*). However, ascorbate plays an important role in protecting photosynthetic cells against ROS derived from the chloroplast (*Smirnoff, 2011*). Ascorbate peroxidase (APX), which is found in both the cytosol and the chloroplast of photosynthetic eukaryotes, is central to this photoprotective role. Thylakoid- and stroma-localised APX removes $H_2O_2$ produced by photosystem I through the activity of the ascorbate-glutathione cycle and this process may account for 10% of photosynthetic electron transport flow. Ascorbate also plays a critical role in preventing lipid peroxidation in the thylakoid membranes and acts as a co-factor for violaxanthin de-epoxidase in the xanthophyll cycle. In addition, ascorbate in the thylakoid lumen may prevent photoinhibition in high light by directly donating electrons to the photosynthetic electron transport chain (*Smirnoff, 2011*). These roles have been demonstrated in a range of ascorbate deficient plants that display sensitivity to high light and to oxidants (*Smirnoff, 2011*).

Photosynthetic eukaryotes arose following the endosymbiotic acquisition of a cyanobacterial ancestor by a non-photosynthetic eukaryote in the Archaeplastida (Plantae) lineage (*Keeling, 2010*). Several other eukaryote lineages, including the diatoms, haptophytes and euglenids, subsequently

**eLife digest** Animals, plants, algae and other eukaryotic organisms all need vitamin C to enable many of their enzymes to work properly. Vitamin C also protects plant and algal cells from damage by molecules called reactive oxygen species (ROS), which can be produced when these cells harvest energy from sunlight in a process called photosynthesis. Photosynthesis occurs inside structures called chloroplasts, and has evolved on multiple occasions in eukaryotes when non-photosynthetic organisms acquired chloroplasts from other algae and then had to develop improved defences against ROS.

There are several steps involved in the production of vitamin C. In many animals, an enzyme called GULO carries out the final step by converting a molecule known as an aldonolactone into vitamin C; this reaction also produces ROS as a waste product. The GULO enzyme is missing in humans, primates and some other groups of animals, so these organisms must get all the vitamin C they need from their diet.

Plants and algae use a different enzyme—called GLDH—to make vitamin C from aldonolactone. GLDH is very similar to GULO, but it does not produce ROS as a waste product. It is not clear how the different pathways have evolved, or why some animals have lost the ability to make their own vitamin C.

Here, Wheeler et al. used genetics and biochemistry to investigate the evolutionary origins of vitamin C production in a variety of eukaryotic organisms. This investigation revealed that although GULO is missing from the insects and several other groups of animals, it is present in the sponges and many other eukaryotes. This suggests that GULO evolved in early eukaryotic organisms and has since been lost by the different groups of animals. On the other hand, GLDH is only found in plants and the other eukaryotes that can photosynthesize.

Wheeler et al.'s findings suggest that GULO has been lost and replaced by GLDH in all plants and algae following their acquisition of chloroplasts. GDLH allows plants and algae to make vitamin C without also producing ROS, which could explain why vitamin C has been able to take on an extra role in these organisms. The results allow us to better understand the functions of vitamin C in photosynthetic organisms and the processes associated with the acquisition of chloroplasts during evolution.

gained plastids through a secondary endosymbiosis with either a red or green alga. These plastid endosymbioses were accompanied by lateral gene transfer on a massive scale from the symbiont to the host nuclear genome (known as endosymbiotic gene transfer or EGT), giving rise to the complex physiologies of photosynthetic eukaryotes (*Timmis et al., 2004*). The plastids in the major photosynthetic eukaryote lineages are all ultimately derived from the primary endosymbiosis. Whilst acquisition of a photosynthetic endosymbiont may have been beneficial to the host cell in many ways, the plastid is also a major source of potentially damaging ROS (*Dorrell and Howe, 2012*). There is evidence for extensive leakage of $H_2O_2$ out of plastids via aquaporins, particularly at high light intensities (*Mubarakshina et al., 2010*; *Naydov et al., 2012*). Plastid acquisition is therefore associated with a greatly increased requirement for cellular antioxidant systems to prevent photodamage. Cyanobacteria do not possess APX or any of the known enzymes for ascorbate biosynthesis and minimise photo-oxidative stress using alternative mechanisms, such as peroxiredoxins, catalases and glutathione peroxidases (*Zámocký et al., 2010*; *Gest et al., 2013*). This suggests that ascorbate was most likely recruited to its photoprotective role after the acquisition of the plastid in ancestral Archaeplastida (*Gest et al., 2013*), although the evolutionary origins of ascorbate biosynthesis are unclear.

There is no clear evidence for ascorbate biosynthesis in prokaryotes (see 'Materials and methods'), suggesting that the ability to synthesise ascorbate evolved in eukaryotes. The three eukaryote lineages in which ascorbate biosynthesis has been examined extensively (animals, plants and *Euglena*) all exhibit different biosynthetic pathways (*Figure 1*). These pathways may have arisen due to convergent evolution, or may represent modifications of an ancestral pathway. An understanding of these evolutionary relationships will provide insight into the cellular roles of ascorbate in eukaryotes, particularly in relation to plastid acquisition in the photosynthetic lineages.

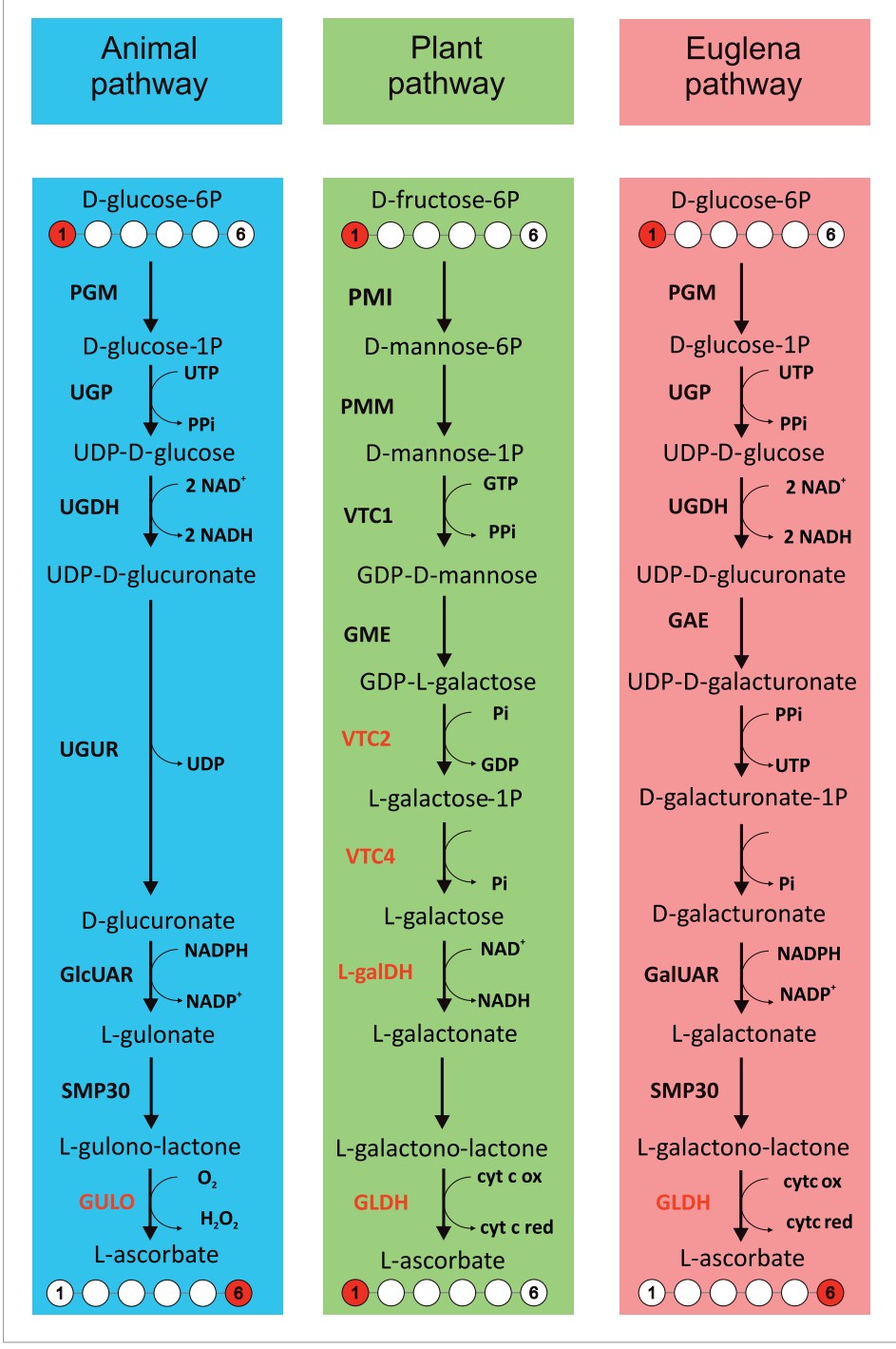

**Figure 1**. Major ascorbate biosynthetic pathways in eukaryotes. The scheme depicts the three major ascorbate biosynthetic pathways found in eukaryotes (*Shigeoka et al., 1979*; *Wheeler et al., 1998*; *Linster and Van Schaftingen, 2007*). The plant pathway (also known as the Smirnoff-Wheeler or D-mannose/L-galactose pathway) involves no inversion of the carbon chain (i.e., C1 of D-glucose becomes C1 of L-ascorbate), whereas the euglenid and animal pathways involve inversion of the carbon chain in the conversion from uronic acid to aldonolactone (i.e., C1 of D-glucose becomes C6 of L-ascorbate). Our analyses focus on enzymes with a dedicated role in ascorbate biosynthesis (shown in red): GULO—L-GulL oxidase; VTC2—GDP-L-galactose phosphorylase; VTC4—L-galactose-1-phosphate phosphatase; L-galDH—L-galactose dehydrogenase; GLDH—L-GalL dehydrogenase. The other enzymes are: PGM—phosphoglucomutase; UGP—UDP-D-glucose pyrophosphorylase; UGDH—UDP-D-glucose

*Figure 1. continued on next page*

Figure 1. Continued

dehydrogenase; UGUR—UDP-glucuronidase; GlcUAR—D-glucuronate reductase; SMP30—regucalcin/lactonase; GAE—UDP-D-glucuronate-4-epimerase; GalUAR—D-galacturonate reductase. Enzyme names are not listed for steps where multiple enzymes may be involved or where specific enzymes have not been identified.

Animals synthesise ascorbate via D-glucuronic acid and L-gulonolactone (L-GulL), with L-gulonolactone oxidase (GULO) catalysing the oxidation of L-GulL to ascorbate. Many animal lineages have lost the ability to synthesise ascorbate, including haplorhine primates, guinea pigs, teleost fish, some bats and passerine birds. In all of the animal ascorbate auxotrophs that have been examined, the gene encoding GULO is lost (*Nishikimi et al., 1992, 1994*; *Cui et al., 2011a*; *Drouin et al., 2011*). GULO uses molecular $O_2$ as its electron acceptor, resulting in $H_2O_2$ production, which may have contributed to selective pressure to lose this enzyme in some animals (*Bánhegyi et al., 1996*; *Mandl et al., 2009*). The inability of many invertebrates to synthesise ascorbate led to early speculation that ascorbate synthesis may have evolved later in metazoan evolution (*Chatterjee, 1973*). However, whilst the loss of *GULO* in vertebrate lineages has been extensively examined (*Yang, 2013*), little is known about its distribution in invertebrates or the non-metazoan members of the holozoa. This information is required to identify the selective pressures underlying the evolution of ascorbate auxotrophy in animals.

Two alternative routes to ascorbate biosynthesis have been identified in photosynthetic eukaryotes, which both employ L-galactonolactone dehydrogenase (GLDH) as the terminal enzyme instead of GULO. A pathway via D-galacturonic acid and L-galactonolactone (L-GalL) was identified in *Euglena* (*Shigeoka et al., 1979*). This pathway is analogous to the animal pathway and also appears to be functional in some stramenopile algae (*Helsper et al., 1982*; *Grün and Loewus, 1984*). In contrast, ascorbate biosynthesis in land plants was found to occur via a different route using D-mannose and L-galactose (*Wheeler et al., 1998*). Green algae also use the 'plant pathway' (*Running et al., 2003*; *Urzica et al., 2012*), but evidence is lacking for the nature of ascorbate biosynthesis in many other evolutionarily important lineages, most notably the rhodophytes (red algae).

This paper focuses on the distribution of the three major pathways of ascorbate biosynthesis described above. Alternative routes of ascorbate biosynthesis have been described in trypanosomes and also in the fungi, which synthesise a range of ascorbate analogues (see 'Materials and methods') (*Loewus, 1999*; *Logan et al., 2007*). There is some evidence for the operation of alternative routes to ascorbate in land plants (*Wolucka and Van Montagu, 2003*; *Lorence et al., 2004*; *Badejo et al., 2012*), although molecular genetic evidence from *Arabidopsis* indicates that the D-mannose/L-galactose pathway is the primary route of ascorbate biosynthesis (see 'Materials and methods') (*Conklin et al., 1999*; *Dowdle et al., 2007*).

The three major pathways of ascorbate biosynthesis therefore all utilise different routes to synthesise an aldonolactone precursor (L-gulonolactone, L-GulL or L-galactonolactone, L-GalL), which is converted to ascorbate by either GULO (animal pathway) or GLDH (plant and euglenid pathways) (*Shigeoka et al., 1979*; *Wheeler et al., 1998*; *Loewus, 1999*). GULO and GLDH exhibit significant sequence similarity and are both members of the vanillyl alcohol oxidase (VAO) family of flavoproteins (*Leferink et al., 2008*). These similar enzymes exhibit important biochemical differences. GULO can oxidise L-GulL and L-GalL, whereas GLDH is highly specific for L-GalL (*Smirnoff, 2001*). GULO localises to the lumen of the endoplasmic reticulum (ER), whereas GLDH is associated with complex I in the mitochondrial electron transport chain (*Schertl et al., 2012*). Importantly, GLDH does not generate $H_2O_2$, as it uses cytochrome *c* rather than $O_2$ as an electron acceptor.

Despite the importance of ascorbate in eukaryote physiology, it is not known how the different pathways of ascorbate biosynthesis arose in animals, plants and algae or relate to its differing cellular roles. This manuscript examines the origins of ascorbate biosynthesis in eukaryotes and seeks to address the following important gaps in our current knowledge: (1) what is the wider distribution of *GULO* loss and ascorbate auxotrophy in the metazoa? (2) do all photosynthetic eukaryotes use an alternative terminal enzyme to animals? (3) why do two different pathways using GLDH exist in photosynthetic eukaryotes? (4) which pathway is used in the rhodophytes? Using a combination of molecular and biochemical analyses, we present evidence that *GULO* is an ancestral gene in eukaryotes that has been functionally replaced by *GLDH* in the photosynthetic lineages, resulting in the development of their alternative biosynthetic pathways.

## Results

### Distribution of GULO and GLDH in eukaryote genomes

To examine the origins of ascorbate biosynthesis in eukaryotes we analysed the distribution of *GULO* and *GLDH* in eukaryote genomes. We found that *GULO* and *GLDH* have a mutually exclusive distribution (*Figure 2*). *GULO* is absent from many metazoan genomes, including all insects, supporting earlier biochemical evidence that insects are predominately ascorbate auxotrophs (*Supplementary file 1*) (*Chatterjee, 1973*; *Dadd, 1973*). However, *GULO* is present in basally derived metazoans, including sponges and cnidarians, and is also present in a filasterean (*Capsaspora owczarzaki*) and in fungi (*Supplementary file 1*). This suggests that ascorbate synthesis via GULO is an ancestral trait in the Opisthokonta that has been lost in many lineages.

*GULO* is also present the Apusomonadida (*Thecamonas trahens*), a sister group to the Opisthokonts, and in members of the Amoebozoa and Excavata. Surprisingly, we also found *GULO* in basally derived Archaeplastida including the glaucophyte, *Cyanophora paradoxa*, and the rhodophytes *Galdieria sulphuraria* and *Galdieria phlegrea*. The glaucophytes occupy a key position in the evolution of photosynthetic eukaryotes as they diverged from the other Archaeplastida before the split of the red and green algal lineages and have highly unusual chloroplasts (termed cyanelles) that retain several features of the cyanobacterial endosymbiont (*Price et al., 2012*). *GULO* is absent from all other Archaeplastida genomes, although an enzyme family exhibiting weak similarity to *GULO* has been reported in *Arabidopsis* (*Maruta et al., 2010*). However, this enzyme forms a distinct phylogenetic clade from all other *GULO* and *GLDH* sequences and its role in de novo ascorbate biosynthesis remains unclear.

*GLDH* was found in all Archaeplastida genomes, except for *Cyanophora* and *Galdieria*, and in all photosynthetic lineages that have acquired a plastid via secondary endosymbiosis (including stramenopiles, cryptophytes, haptophytes, chlorarachniophytes and euglenids). *GLDH* was present in several non-photosynthetic organisms including the oomycetes, the foraminifera and in the choanoflagellates, *Monosiga brevicollis* and *Salpingoeca rosetta*. The evolutionary history of algal plastids acquired by secondary endosymbiosis remains uncertain and there is some evidence that non-photosynthetic stramenopile (e.g., oomycetes) and rhizarian (e.g., foraminifera) lineages may have once acquired a plastid that was subsequently lost (*Tyler et al., 2006*; *Keeling, 2010*; *Glöckner et al., 2014*).

Further identification of *GLDH* or *GULO* in the transcriptomes of 165 eukaryotes within the Marine Microbial Eukaryote Transcriptome dataset (*Keeling et al., 2014*) confirmed that *GLDH* was found primarily in photosynthetic organisms (*Supplementary file 2*), but also in the non-photosynthetic stramenopiles such as oomycetes, biocosoecids and labyinthulids and in an acanthoecid choano-flagellate. The presence of *GULO* was restricted to non-photosynthetic organisms, including the heterotrophic flagellate *Palpitomonas bilix*, which is a non-photosynthetic relative of the Cryptophyte algae (*Yabuki et al., 2014*). The exception was the presence of *GULO* in the chromerids, *Chromera velia* and *Vitrella brassicaformis*, which are photosynthetic relatives of the Apicomplexa (*Janouskovec et al., 2010*).

Further searches of Expressed Sequence Tag (EST) and Transcriptome Shotgun Assembly (TSA) databases identified a *GULO* sequence in the green alga, *Chlorokybus atmophyticus* (JO192417.1) (*Leliaert et al., 2012*; *Timme et al., 2012*). *Chlorokybus* represents a basal lineage in the charophyte algae, which are a sister group to the land plants (*Figure 2—figure supplement 1*). *GLDH* was identified in the other charophytes *Klebsormidium flaccidum*, *Nitella mirabilis* and *Nitella hyalina* (JO285109.1, JV744884.1, JO253095.1). These searches also revealed the presence of *GULO* in the craspedid choanoflagellate, *Monosiga ovata* (DC478225.1). The Craspedida subgroup of the choanoflagellates is divided into two major clades; clade I contains *M. brevicollis* and *S. rosetta* (which both possess *GLDH*) and clade II contains *M. ovata* (*Jeuck et al., 2014*).

We conclude that nearly all photosynthetic eukaryotes use *GLDH* rather than *GULO* as the terminal enzyme in ascorbate biosynthesis. The exceptions are the basally derived Archaeplastida (*Cyanophora*, *Galdieria* and *Chlorokybus*) and the chromerids (*Figure 2—figure supplement 2*). No photosynthetic organisms were found to lack both *GULO* and *GLDH*, although both genes were absent in many non-photosynthetic organisms, including all insects, *Daphnia*, *Paramecium* and *Dictyostelium*. The genomes of many parasitic groups also appear to lack both enzymes including the

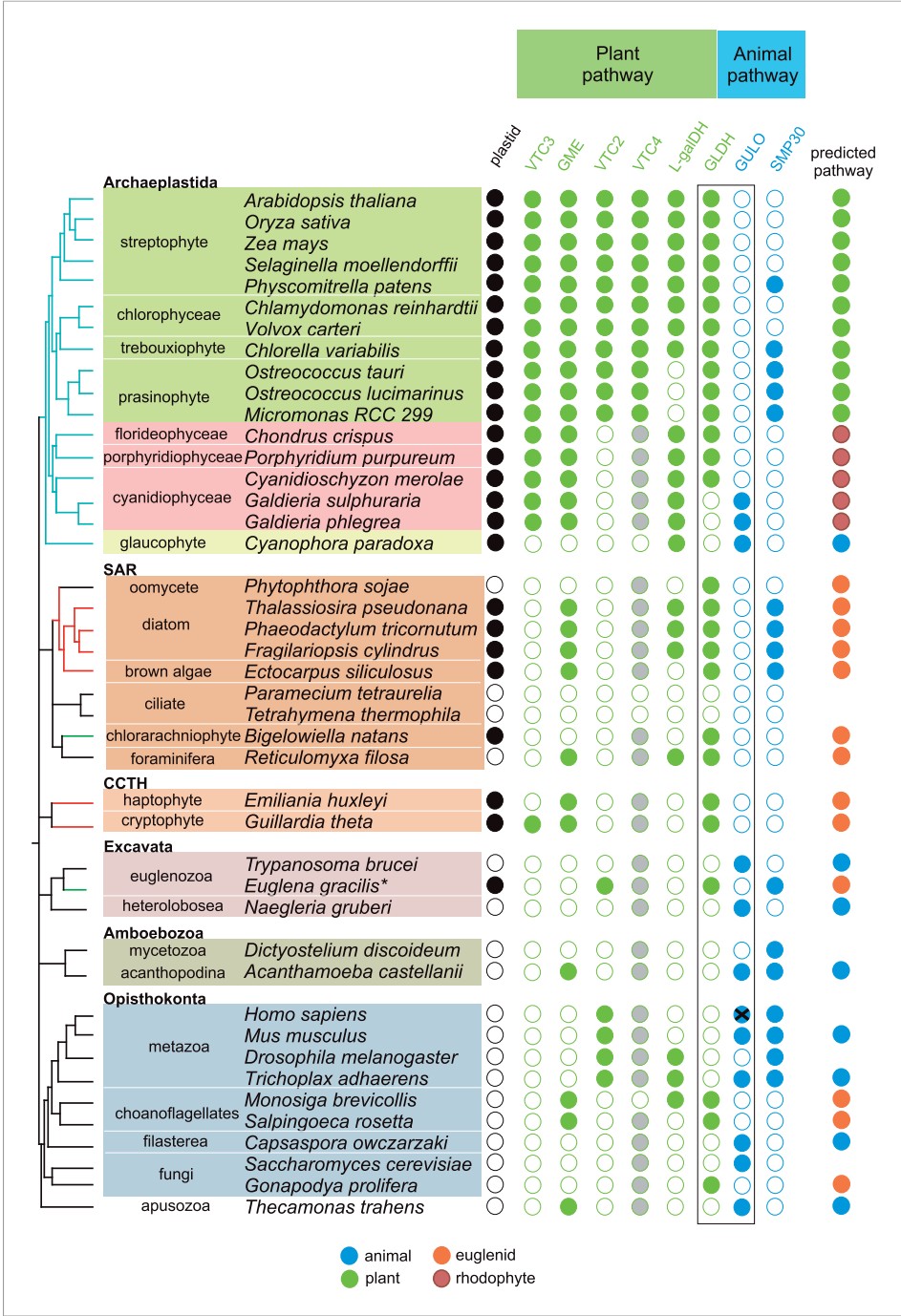

**Figure 2**. Coulson plot indicating the taxonomic distribution of the different ascorbate pathways. 40 eukaryote genomes were analysed for the presence of genes in the ascorbate biosynthetic pathways. The two potential terminal enzymes in the pathway are boxed. *GLDH* is common to both the 'plant' and 'euglenid' type pathways. A schematic tree depicts the currently accepted phylogenetic relationships between organisms. The predicted route of ascorbate biosynthesis in each organism is shown. Note that 'euglenid' and 'rhodophyte' type pathways cannot currently be distinguished from sequence analysis alone and the predictions are based on biochemical evidence. Asterisk denotes a genome assembly was not available for *Euglena gracilis* and its transcriptome was analysed ('Materials and methods'). Grey circles in VTC4 represent the presence of a highly similar enzyme, *myo*-inositol-1-phosphate phosphatase that exhibits L-galactose-1-phosphatase activity. GULO in trypanosomes and yeasts acts to oxidise the alternative substrates L-galactonolactone or D-arabinonolactone respectively. VTC3 is not a biosynthetic enzyme, but represents a dual function Ser/Thr protein kinase/protein phosphatase 2C that may play a regulatory role in the plant pathway (**Conklin et al., 2013**). Black cross represents a pseudogene encoding a non-functional

*Figure 2. continued on next page*

*Figure 2. Continued*

enzyme. Organisms with sequenced genomes that were found to lack both of the terminal enzymes in the known pathways (GULO and GLDH) are likely to be ascorbate auxotrophs and were not included in the plot. These include *Giardia intestinalis*, *Trichomonas vaginalis*, *Entamoeba invadens*, *Plasmodium falciparum* and *Perkinsus marinus*.

The following figure supplements are available for figure 2:

**Figure supplement 1**. Distribution of GULO and GLDH in the Archaeplastida.

**Figure supplement 2**. Distribution of the different ascorbate pathways.

---

diplomonads, parabasalids and apicomplexa (e.g., *Giardia intestinalis*, *Trichomonas vaginalis* and *Plasmodium falciparum*). Since all documented pathways of ascorbate biosynthesis require either GULO or GLDH, organisms lacking both of these enzymes are likely to be ascorbate auxotrophs (*Chatterjee, 1973*).

## Phylogenetic analyses of GULO and GLDH

A maximum likelihood tree of *GULO* and *GLDH* sequences was produced using the other members of the VAO family as an outgroup to root the tree. *GLDH* is highly conserved and the phylogenetic analyses strongly support a monophyletic origin for all *GLDH* sequences (100% bootstrap support, posterior probability = 1) (*Figure 3*). The monophyly of eukaryote *GULO* sequences is well supported (85% bootstrap support, posterior probability = 1). This clade includes trypanosome L-GalL oxidase and ascomycete D-arabinonolactone oxidase (*Huh et al., 1994*; *Logan et al., 2007*), indicating that although these enzymes exhibit altered substrate specificity they should be considered within the *GULO* clade for our evolutionary analyses. There is also moderate support for the monophyly of *GULO* and *GLDH* as a clade within the VAO family, supporting hypotheses that these enzymes may have originated from a gene duplication event. GLDH is highly unusual amongst the VAO flavoprotein family in that it does not use $O_2$ as an electron acceptor, but mutation of a single highly conserved alanine residue in GLDH is required to convert it from a dehydrogenase to an oxidase (*Leferink et al., 2009*). The phylogenies within the *GULO* clade and the *GLDH* clade are poorly resolved, and so the trees do not provide evidence on the likelihood of lateral gene transfer, such as endosymbiotic gene transfer (EGT) or horizontal gene transfer (HGT), of either gene. Further phylogenetic analyses, individually examining each gene using unrooted trees with much greater taxonomic sampling, were unable to provide greater resolution (*Figure 3—figure supplement 1*).

## Distribution of GLDH-dependent pathways in photosynthetic organisms

Many of the enzymes preceding GULO or GLDH in the animal and euglenid pathways play other roles within the cell, for example, in uronic acid metabolism or providing pentose intermediates (*Linster and Van Schaftingen, 2007*). The presence or absence of these genes is therefore not solely related to ascorbate biosynthesis. However, the plant pathway of ascorbate biosynthesis contains a number of dedicated enzyme steps, allowing a much clearer examination of its distribution. This also enables a distinction to be made between the plant- and euglenid-type pathways, as both utilise GLDH as the terminal enzyme. Plants and green algae use GDP-L-galactose phosphorylase (VTC2) and L-galactose dehydrogenase to generate L-GalL (*Gatzek et al., 2002*; *Running et al., 2003*; *Dowdle et al., 2007*; *Laing et al., 2007*; *Linster et al., 2007*), whereas euglenids use D-galacturonate reductase (*Figure 1*) (*Ishikawa et al., 2006*). We found that L-galactose dehydrogenase is present in all rhodophytes and Viridiplantae, except the prasinophytes *Ostreococcus* and *Micromonas* (see 'Materials and methods'). Sequences exhibiting similarity to L-galactose dehydrogenase were also found in the diatoms and in some metazoa, but as some of these species are ascorbate auxotrophs, it appears that this enzyme may play alternative metabolic roles. *VTC2* is found exclusively in the Viridiplantae (including *Chlorokybus*), indicating that the definitive 'plant' pathway is restricted to this lineage (*Figure 2*; *Supplementary files 3, 4*). Biochemical evidence from euglenids and stramenopiles (*Shigeoka et al., 1979*; *Helsper et al., 1982*; *Grun and Loewus, 1984*) suggests that organisms that lack *VTC2* but possess *GLDH* are likely to operate a 'euglenid' pathway, with D-galacturonic acid acting as the precursor of L-GalL. However, it is not clear whether this is also the case in the rhodophytes. All

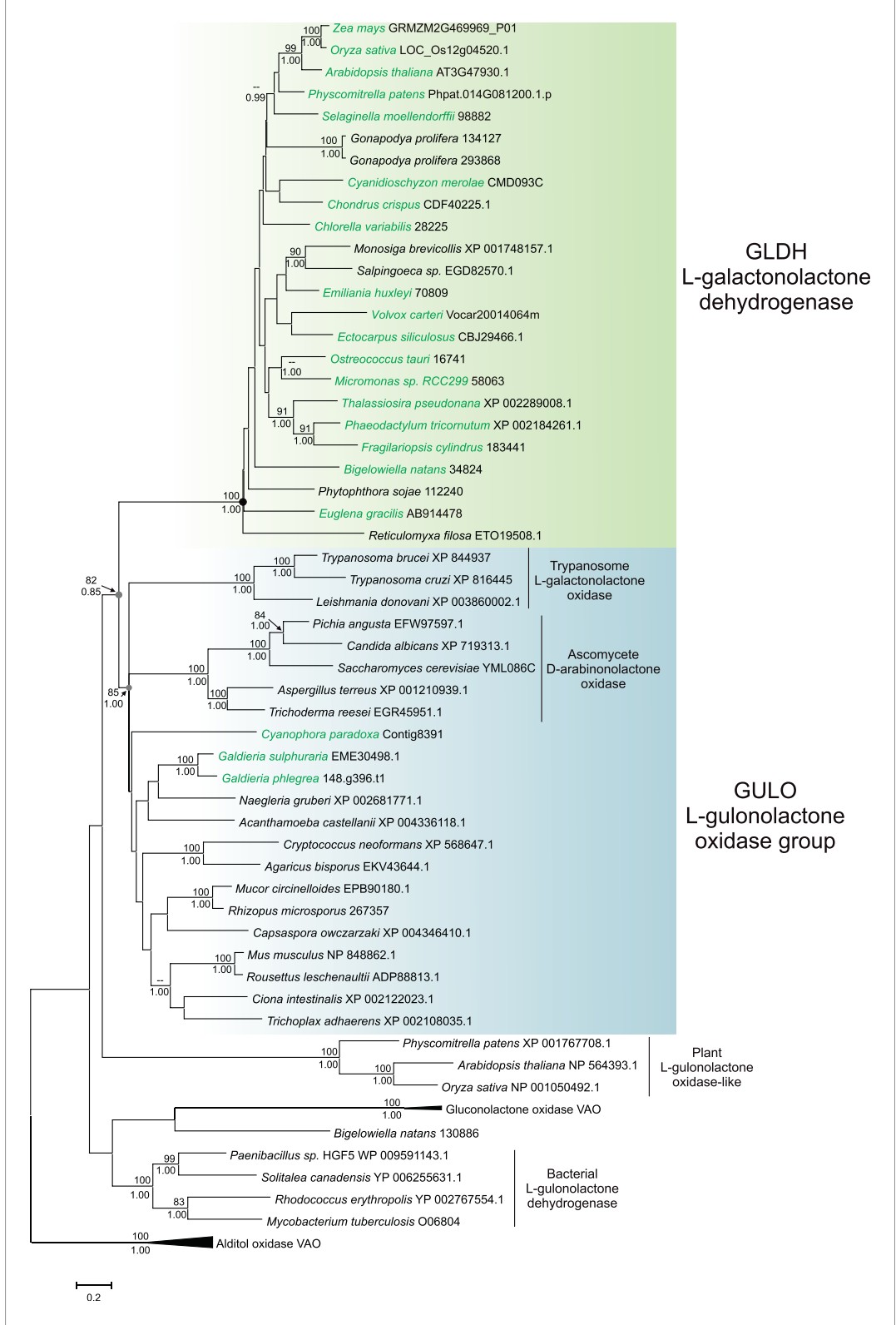

**Figure 3**. Phylogenetic analysis of L-gulonolactone oxidase and L-galactonolactone dehydrogenase. A maximum likelihood phylogenetic tree demonstrating the relationships between aldonolactone oxidoreductases involved in ascorbate biosynthesis. A multiple sequence alignment of 263 amino acid residues was used with alditol oxidases from the vanillyl alcohol oxidase (VAO) family acting as the outgroup. Photosynthetic organisms are shown in green. *Figure 3. continued on next page*

Figure 3. Continued
There is strong support for a monophyletic origin for GLDH in eukaryotes. Bootstrap values >80% are shown above nodes (100 bootstraps) and Bayesian posterior probabilities >0.95 are shown below (10000000 generations), except for selected key nodes (circled) where all values are displayed.
The following figure supplement is available for figure 3:

**Figure supplement 1**. Phylogenetic analysis of L-galactonolactone dehydrogenase.

rhodophytes possess a sequence that is highly similar to characterised L-galactose dehydrogenases from plants and bacteria (*Gatzek et al., 2002*; *Hobbs et al., 2014*) and L-galactose residues are a major constituent of red algal polysaccharides (*Percival, 1979*). We therefore examined whether rhodophytes could synthesise ascorbate via a modified 'plant' pathway using an alternative route to L-galactose.

## Biochemical analysis of pathways in rhodophytes

We used the macroalga, *Porphyra umbilicalis*, to examine whether rhodophytes can utilise L-galactose in ascorbate synthesis. We detected $NAD^+$-dependent L-galactose dehydrogenase activity in *Porphyra* thallus extracts (*Figure 4A*). Feeding 10 mM L-galactose to *Porphyra* thallus slices increased the concentration of ascorbate (detected as dehydroascorbate by GC-MS) (*Figure 4B*). As red algae synthesise GDP-L-galactose from GDP-D-mannose (*Su and Hassid, 1962*), rhodophytes likely use a modified 'plant pathway' to synthesise ascorbate, employing an unidentified enzyme activity to generate L-galactose from GDP-L-galactose instead of VTC2.

We then examined ascorbate biosynthesis in *Galdieria*, which differs from all other rhodophytes (*Supplementary file 4*) in that it possesses *GULO* rather than *GLDH*. L-galactose, L-GalL and L-GulL were all effective precursors of ascorbate in *G. sulphuraria* (*Figure 4C*), suggesting that L-galactose is converted to L-GalL, which may then be converted to ascorbate by GULO. A positional isotopic labelling approach indicated that label from D-[1-$^{13}$C]-glucose was incorporated primarily into carbon 1 (C1) of ascorbate (*Figure 4D*, *Figure 4—figure supplement 1*). This labelling pattern is expected for the plant pathway, while the reduction of a uronic acid intermediate in the animal or euglenid pathways would result in the transfer of label from C1 of glucose into C6 of ascorbate/ dehydroascorbate (*Loewus, 1999*). *G. sulphuraria* therefore uses a similar pathway to other rhodophytes, employing GULO instead of GLDH.

In combination, these data identify a clear difference between the Archaeplastida and the photosynthetic lineages that have acquired a plastid via secondary endosymbiosis. Whilst both groups use GLDH as the terminal enzyme for ascorbate synthesis, they differ in the route to L-GalL, The Archaeplastida generate L-GalL via L-galactose (without inversion of the carbon chain of glucose), whereas photosynthetic eukaryotes with secondary plastids synthesise L-GalL via D-galacturonate, resulting in inversion of the carbon chain.

## Distribution of ascorbate-dependent antioxidant systems

We have found that nearly all photosynthetic eukaryotes use GLDH to synthesise ascorbate, suggesting that this distribution may be linked to the photoprotective role of ascorbate. We therefore determined the distribution of ascorbate-dependent antioxidant mechanisms in eukaryote genomes. Three main isoforms of APX are found in eukaryotes (APX, APX-R and APX-CCX, a hybrid enzyme containing both ascorbate and cytochrome *c* peroxidase domains) (*Zamocky et al., 2010*; *Lazzarotto et al., 2011*; *Fawal et al., 2013*). APX and APX-R are found in nearly all photosynthetic eukaryotes and in the choanoflagellates, *M. brevicollis* and *S. rosetta*. *Euglena* and *Emiliania* do not possess APX or APX-R, although both possess APX-CCX, which is also found in some fungi and in *Capsaspora* (*Figure 5*). The only photosynthetic eukaryote in this analysis that lacks any isoform of APX was *C. paradoxa*. *Cyanophora* also lacks all of the remaining enzymes of the plant ascorbate-glutathione cycle: monodehydroascorbate reductase (MDHAR), dehydroascorbate reductase (DHAR) and glutathione reductase (GR). The ascorbate-dependent xanthophyll cycle is not present in glaucophytes and so *Cyanophora* does not require ascorbate for non-photochemical quenching (*Figure 5*). Moreover, the cellular concentration of ascorbate in *Cyanophora* is either very low or absent, as we could not detect ascorbate in *Cyanophora* extracts using GC-MS (data not shown). It is

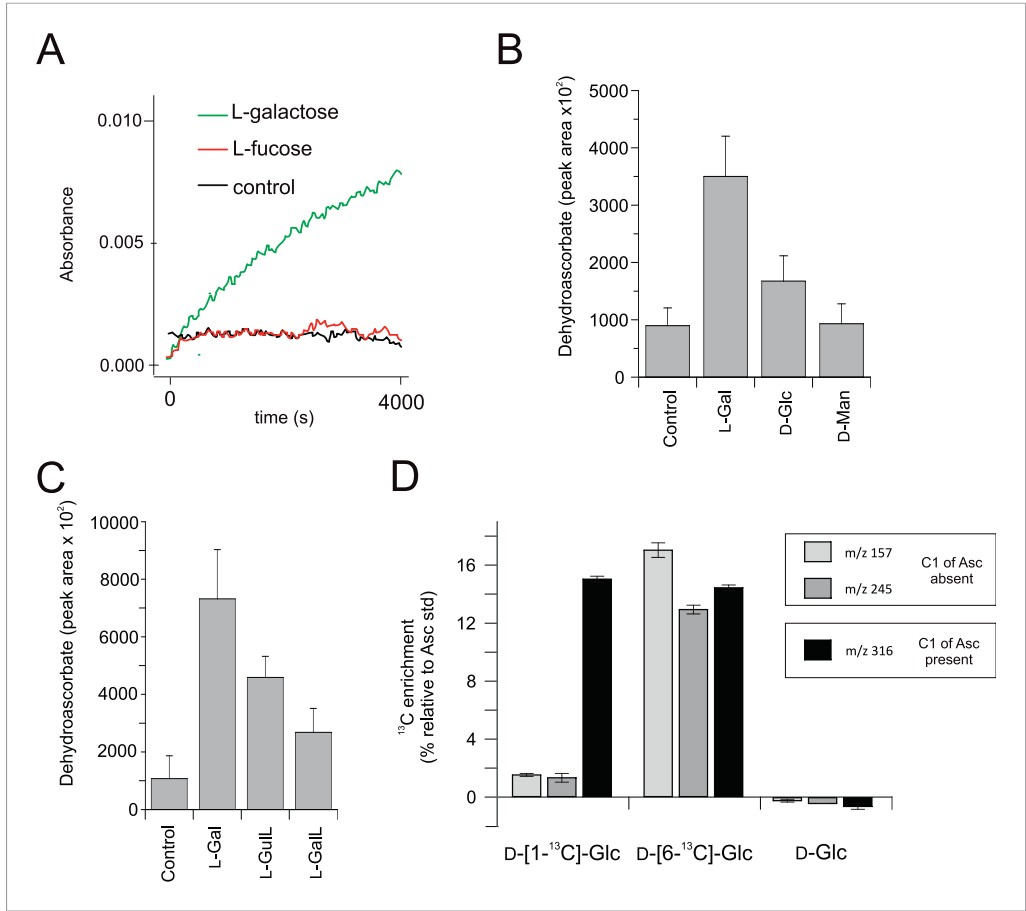

**Figure 4**. Biochemical evidence for a modified D-mannose/L-galactose pathway in rhodophytes. (**A**) Crude extracts of *Porphyra umbilicalis* thallus demonstrate L-galactose dehydrogenase activity using 5 mM L-galactose (L-Gal) as a substrate. No activity was demonstrated with 5 mM L-fucose (6-deoxy-L-galactose) as a substrate. The result is representative of three different enzyme preparations. (**B**) Feeding 10 mM L-Gal to *Porphyra* thallus for 24 hr resulted in an accumulation of ascorbate (detected as dehydroascorbate—DHA). D-mannose (10 mM) did not cause an increase in ascorbate in *Porphyra*, but exogenous D-mannose does not elevate ascorbate in land plants even though it is an intermediate in ascorbate biosynthesis. The bar chart shows mean peak areas of selected fragments (±s.d.). n = 3. (**C**) Feeding ascorbate precursors (25 mM) to *Galdieria sulphararia* from both the plant and animal pathways results in increased cellular ascorbate (detected as dehydroascorbate using GC-MS) (±s.d.). The extent of the increase in cellular ascorbate is influenced by the rate of conversion of the intermediate and the rate of its uptake into the cell. n = 3. (**D**) Feeding D-[1-$^{13}$C]-glucose (25 mM) to *Galdieria sulphararia* results in enrichment of $^{13}$C in the 316/317 m/z fragment of dehydroascorbate (which includes C1), but not in the 245/246 m/z or 157/158 m/z fragment (which exclude C1) (±s.d.). In contrast, feeding D-[6-$^{13}$C]-glucose (25 mM) labels all fragments, suggesting that they all include C6. In combination, this labelling pattern indicates plant-like non-inversion of the carbon chain in the conversion of hexoses to ascorbate. n = 3.

The following figure supplement is available for figure 4:

**Figure supplement 1**. Positional isotopic labelling of ascorbate biosynthesis.

possible that ascorbate analogues are present that we could not identify. However, in combination with the lack of the plant ascorbate-glutathione cycle and the xanthophyll cycle, we conclude that *Cyanophora* is unlikely to rely on ascorbate to detoxify peroxides derived from photosynthesis. *Cyanophora* does however contain several glutathione peroxidases, peroxiredoxins and catalase, as well as a unique peroxidase (symerythrin) similar to rubrerythrin of prokaryotes (*Cooley et al., 2011*). These data suggest that glaucophytes rely on alternative mechanisms to detoxify peroxides derived from photosynthesis. As cyanobacteria also do not appear to use ascorbate for photoprotection

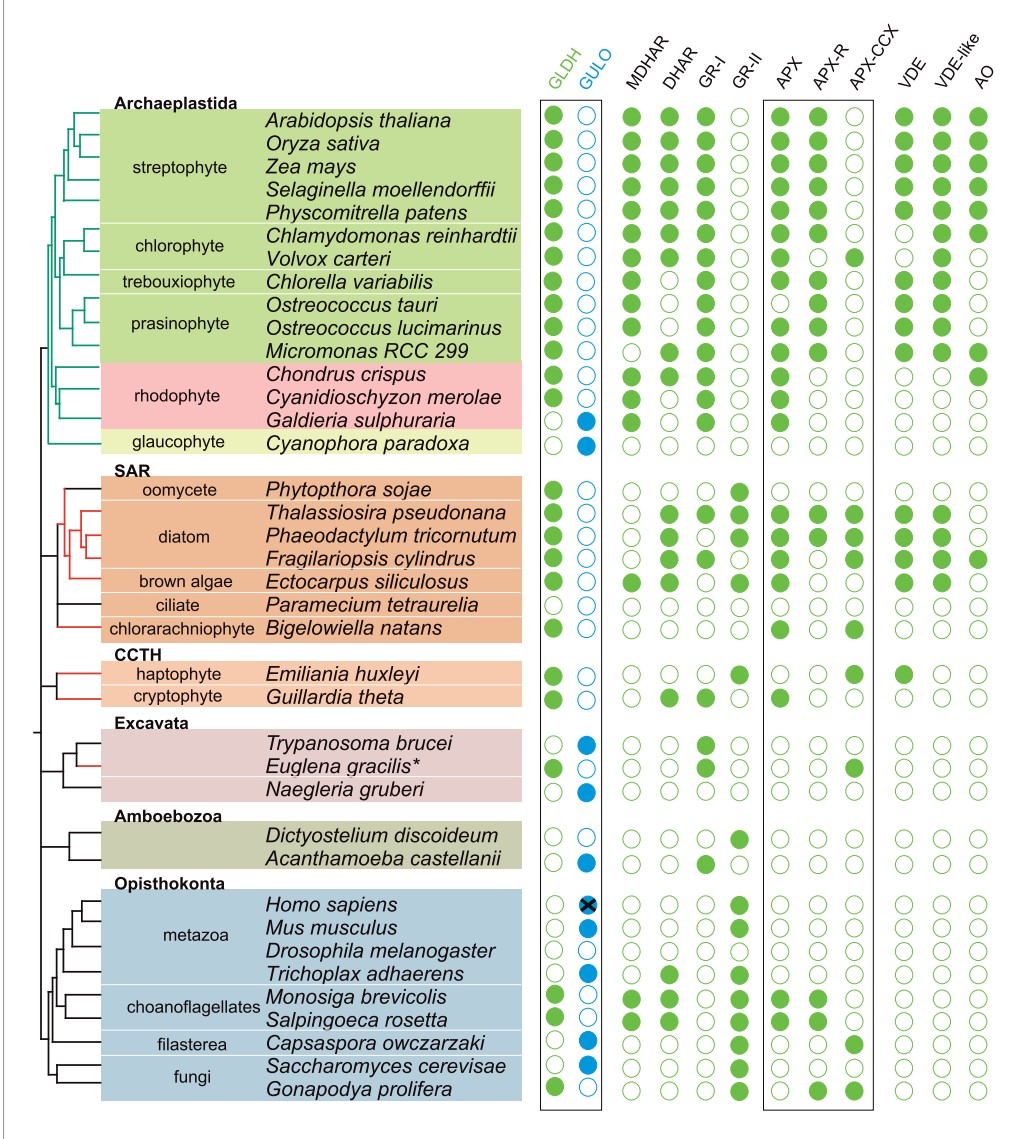

**Figure 5**. Coulson plot showing the distribution of photoprotective ascorbate-dependent enzymes. Eukaryote genomes were analysed for the presence of enzymes from the plant ascorbate-glutathione cycle, the xanthophyll cycle and other ascorbate-dependent enzymes. We found that eukaryotes possess two distinct isoforms of GSH reductase. PeroxiBase was used to distinguish between the different forms of ascorbate peroxidase (*Fawal et al., 2013*). Boxes highlight the terminal enzymes in the biosynthetic pathway and the ascorbate peroxidase family (APX, APX-R and APX-CCX). MDHAR—monodehydroascorbate reductase; DHAR—dehydroascorbate reductase; GR-I—glutathione reductase isoform I; GR-II—glutathione reductase isoform II; APX—ascorbate peroxidase; APX-R—ascorbate peroxidase-related; APX-CCX—hybrid ascorbate peroxidase/cytochrome c peroxidase; VDE—violaxanthin de-epoxidase; VDE-like—violaxanthin de-epoxidase like; AO—ascorbate oxidase.

(*Bernroitner et al., 2009*; *Latifi et al., 2009*), the photoprotective role of ascorbate may therefore have emerged in the Archaeplastida after the divergence of the glaucophytes.

## Discussion

### Distribution of pathways in eukaryotes

Ascorbate (vitamin C) is a very familiar metabolite to humans, so it is perhaps surprising that so many aspects of its biosynthesis and metabolism remain uncharacterised. The biosynthetic pathway of

ascorbate in plants, which supplies the vast majority of ascorbate in the human diet, remained elusive for many years (*Wheeler et al., 1998*) and the major role of ascorbate in DNA demethylation emerged only very recently (*Blaschke et al., 2013*). In order to better understand the cellular roles of ascorbate, we have examined the distribution of the three major pathways of ascorbate biosynthesis in eukaryotes. We identify that the Opisthonkonts (animals and fungi), the Amoebozoa and the non-photosynthetic representatives of the Excavata and CCTH (Hacrobia) use *GULO* for ascorbate biosynthesis. In contrast, the photosynthetic organisms in the Archaeplastida, CCTH (Hacrobia), SAR and the photosynthetic members of the Excavata (euglenids) use *GLDH*. In these photosynthetic organisms, the combination of molecular and biochemical evidence suggests that the non-inversion pathway via L-galactose (plant pathway) is restricted to Archaeplastida, whereas the inversion pathway via D-galacturonate (euglenid pathway) is used by photosynthetic eukaryotes that acquired plastids via secondary endosymbiosis. The important exceptions to these trends are: firstly, that *GLDH* is found in several non-photosynthetic organisms, notably in some choanoflagellates (Opisthokonts) and stramenopiles and secondly, that *GULO* is found in several basally derived members of the Archaeplastida.

## The evolutionary origins of GULO and GLDH

The processes underlying the distribution of the different terminal enzymes are therefore central to our understanding of the evolution of ascorbate biosynthesis. The mutually exclusive distribution of two highly conserved and functionally similar genes in eukaryotes may be explained by either of two evolutionary scenarios: an ancient gene duplication in the last common eukaryote ancestor (ancient paralogy) followed by differential loss of either gene, or lateral gene transfer of a novel gene followed by functional replacement of the ancestral gene (*Keeling and Inagaki, 2004*). It is likely that one of these evolutionary scenarios underlies the distribution of *GULO* and *GLDH* amongst eukaryotes (*Figure 6*).

The model of ancient paralogy requires that both genes were present in the last common eukaryote ancestor, where they both presumably contributed to ascorbate biosynthesis, and were then differentially lost by every eukaryote lineage. This requires that these two functionally similar enzymes co-existed in multiple lineages throughout eukaryote evolution without significant functional divergence. The distribution of *GULO* and *GLDH* in the Archaeplastida and in the choanoflagellates suggests that these enzymes may have coexisted for a time in these lineages (*Figure 2—figure supplement 1*). However, there are no clear examples of extant eukaryotes that possess both enzymes, which would be expected if they have co-existed extensively throughout eukaryote evolution. The ancient paralogy model also requires extensive loss of both *GULO* and *GLDH*. There is clear evidence for multiple independent losses of *GULO* in animals and indications that GULO activity may be deleterious under certain conditions, providing a potential selective pressure for gene loss (*Bánhegyi et al., 1996*; *Hiller et al., 2012*). Similar evidence for the loss of *GLDH* in eukaryotes is lacking. In addition, *Arabidopsis* mutants that lack *GLDH* cannot correctly assemble mitochondrial complex I (*Pineau et al., 2008*), suggesting that loss of *GLDH* is likely to have many wider impacts on metabolism.

The broad distribution of *GULO* supports an ancient evolutionary origin for this gene. It is present in all of the eukaryote supergroups, including basally-derived lineages within the CCTH (Hacrobia) and Archaeplastida and also in the Apusomonads. Although *GLDH* is also present in most of the eukaryote supergroups (except the Amoebozoa), its distribution is primarily restricted to lineages that have acquired a plastid or to isolated lineages (e.g., choanoflagellates). Whilst we cannot discount an ancient origin for *GLDH* in the last common eukaryote ancestor, its distribution may also be reasonably explained by the lateral gene transfer model.

In the lateral gene transfer scenario, either *GULO* or *GLDH* could represent a novel gene that arose in a specific lineage and was then acquired by other eukaryotes through horizontal gene transfer (HGT) or endosymbiotic gene transfer (EGT). However, the distribution of *GULO* cannot be reasonably explained by lateral gene transfer, as this requires HGT on a massive scale specifically into non-photosynthetic eukaryotes. In contrast, the distribution of *GLDH* can be largely explained by EGT during plastid acquisition. *GLDH* may have arisen specifically in ancestral Archaeplastida after the divergence of the glaucophytes and functionally replaced the ancestral gene (*GULO*). *GLDH* could then have been transferred to the other photosynthetic lineages via EGT, resulting in the replacement of *GULO* in lineages that acquired their plastids via secondary endosymbiosis (*Figure 7*). The presence

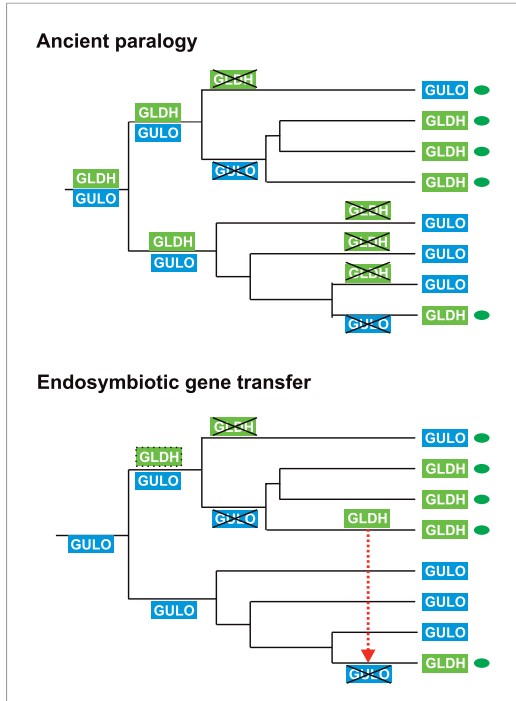

**Figure 6**. Evolutionary scenarios for *GULO* and *GLDH*. The scheme illustrates two most likely evolutionary scenarios responsible for the distribution of *GULO* and *GLDH* in eukaryotes. In the ancient paralogy scenario, an ancient gene duplication in the last common eukaryote ancestor results in the presence of two functionally similar genes, *GULO* and *GLDH*, followed by differential loss of either gene in each lineage. In the endosymbiotic gene transfer (EGT) scenario, *GULO* represents the ancestral gene and *GLDH* represents a novel gene that arose in a specific lineage. EGT of *GLDH* (red dashed arrow) to other photosynthetic lineages (green ovals) enables functional replacement of the ancestral gene. Note that *GULO* represents an ancestral gene in both of these evolutionary scenarios.

of *GLDH* in some non-photosynthetic eukaryotes may be explained by the evolutionary acquisition of a plastid that was subsequently lost. For example, there is some evidence to support plastid loss in non-photosynthetic stramenopiles, although the number and timing of plastid acquisition events via secondary endosymbiosis remains a subject of significant debate (*Keeling, 2010*). The choanoflagellates have not acquired a plastid at any stage, but there is evidence for large scale horizontal gene transfer (HGT) from algae into this lineage, including the HGT of APX (*Nedelcu et al., 2008*). Choanoflagellates may therefore have acquired *GLDH* via HGT along with these other algal genes. EGT of *GLDH* is a more parsimonious scenario than ancient paralogy, as it requires fewer independent loss events. However, the poor resolution of the phylogenies within the *GLDH* clade means that direct evidence for either EGT or HGT between lineages is lacking.

## Evolution of alternative pathways

Both the ancient paralogy and EGT evolutionary scenarios are plausible in the wider context of ascorbate biosynthesis. However, the EGT scenario provides a clear rationale to explain why photosynthetic eukaryotes with primary plastids exhibit a different pathway from those with secondary plastids. Our biochemical evidence suggests that ancestral Archaeplastida developed a non-inversion pathway via L-galactose that employed the broad specificity of GULO to oxidise L-GalL. The development of *GLDH* in ancestral Archaeplastida would have led to the eventual replacement of *GULO* in all red and green algal lineages, except *Galdieria* and *Chlorokybus*, resulting in the non-inversion plant-type pathway found in extant Archaeplas-

tida. In the photosynthetic eukaryotes with secondary plastids, it is likely that the host initially synthesised ascorbate via an animal-type pathway (involving inversion of chain and GULO) and that the red or green algal symbiont used a plant-type pathway (involving non-inversion of the carbon chain and GLDH). However, neither pathway appears to operate in photosynthetic eukaryotes with secondary plastids, which instead use a euglenid-type pathway. We propose that EGT of *GLDH* from the symbiont could have resulted in functional replacement of *GULO* in the animal-type pathway of the host, leading to a hybrid biosynthetic pathway that employed D-galacturonate rather than D-glucuronate as an intermediate in order to provide L-GalL as a substrate for GLDH. The hybrid pathway therefore involves inversion of the carbon chain of D-glucose and GLDH. The generation of a hybrid pathway suggests that photosynthetic eukaryotes with secondary plastids only acquired *GLDH* by EGT rather than the entire plant pathway.

In conclusion, the distribution of *GULO* and *GLDH* in eukaryotes may be explained by either of two evolutionary models; ancient paralogy followed by differential gene loss or EGT of *GLDH* followed by *GULO* loss. We favour the EGT scenario as the most parsimonious and the most consistent with the biochemical evidence, but we cannot rule out either scenario based on the current evidence. Therefore our evolutionary analyses do not allow us to definitively identify the origin of *GLDH*.

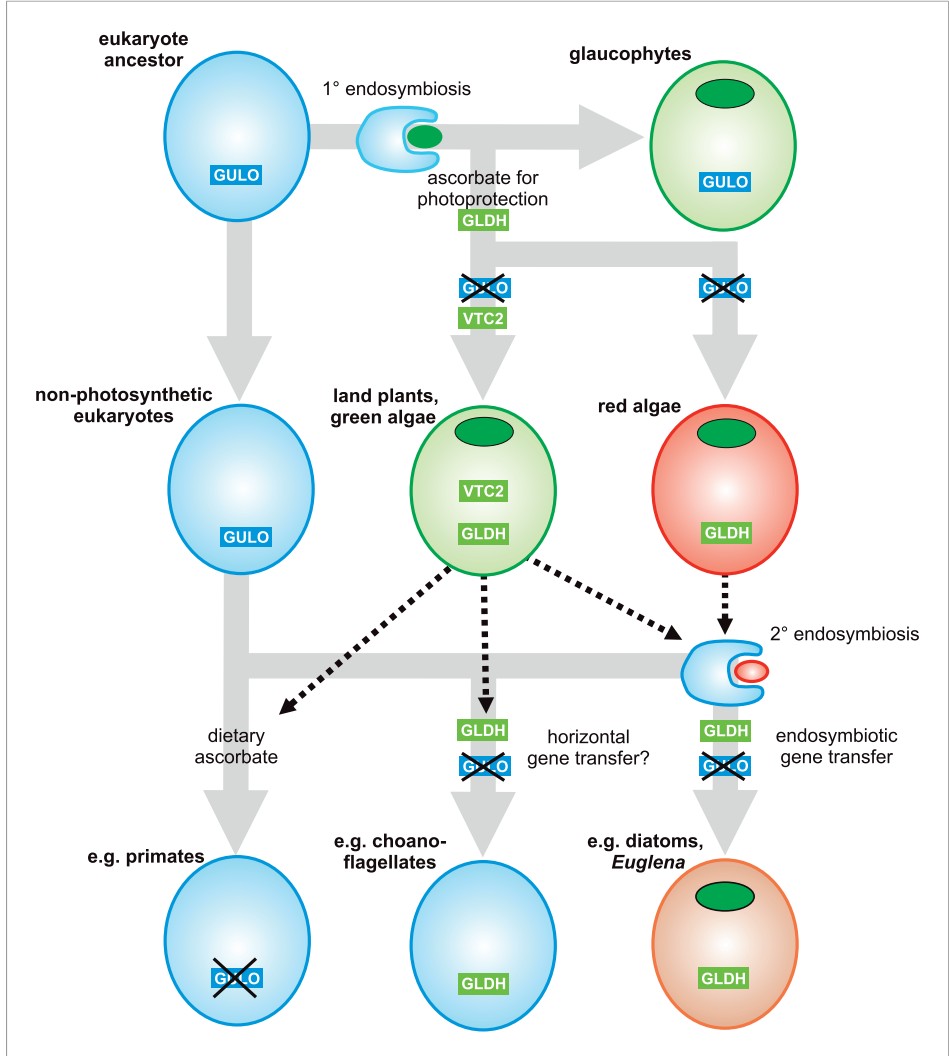

**Figure 7**. A proposed evolutionary model of ascorbate biosynthesis. The scheme illustrates the proposed events in the EGT evolutionary model of eukaryote ascorbate biosynthesis. In this scenario, ancestral eukaryotes synthesised ascorbate via GULO. *GLDH* arose in the Archaeplastida following primary endosymbiosis of a cyanobacterium, after the divergence of the glaucophyte lineage. *GLDH* functionally replaced *GULO* in the red and green algal lineages, coinciding with the rise of the photoprotective role of ascorbate. Plastid acquisition via secondary endosymbiosis of either a green or red alga resulted in endosymbiotic gene transfer of *GLDH* and replacement of *GULO*. As these organisms became the dominant primary producers in many ecosystems, a series of trophic interactions (dotted lines) resulted in the loss of *GULO* in non-photosynthetic organisms, either by providing a ready supply of dietary ascorbate (resulting in ascorbate auxotrophy in heterotrophic organisms) or through putative horizontal gene transfer of *GLDH* (e.g., choanoflagellates). For clarity, not all potential trophic interactions are shown.

However, they do enable clear conclusions to be made on the loss of *GULO*. Both evolutionary models support *GULO* as an ancestral gene in the last common eukaryote ancestor, indicating that *GULO* has been lost in almost all photosynthetic eukaryotes. Therefore, we can conclude that photosynthetic eukaryotes encountered strong selective pressure to replace the function of *GULO* in ascorbate biosynthesis.

## Selective pressures underlying evolution of ascorbate biosynthesis

The critical role of ascorbate in photoprotection has been demonstrated in a diversity of photosynthetic eukaryotes, including land plants, green algae, diatoms and euglenids (*Lavaud and Kroth, 2006*; *Ishikawa et al., 2010*; *Mellado et al., 2012*; *Urzica et al., 2012*). Our analyses indicate

that many photosynthetic eukaryotes possessed GULO prior to plastid acquisition, whereas almost all extant photosynthetic lineages use GLDH to synthesise ascorbate. The selective pressure to replace GULO in ascorbate biosynthesis following plastid acquisition could therefore be linked to the photoprotective role of ascorbate. One intriguing possibility is that the production of $H_2O_2$ by GULO may have limited the ability of the host cell to protect itself against ROS derived from the chloroplast.

Ancestral eukaryotes developed multiple antioxidant mechanisms to protect themselves from ROS derived from organelles such as the peroxisome and the mitochondria. However, the acquisition of a photosynthetic cyanobacterial endosymbiont in the Archaeplastida would have resulted in a greatly increased requirement for cellular antioxidants to protect the host cell from $H_2O_2$ secreted by the plastid. Ascorbate, synthesised in the host cell but not in the cyanobacterial endosymbiont, appears to have been recruited to this role after the divergence of the glaucophytes. The recruitment of ascorbate as a major cellular antioxidant in photosynthetic eukaryotes may have led to an increased requirement for ascorbate biosynthesis. However, ascorbate biosynthesis via GULO results in the production of $H_2O_2$ in the ER lumen. In mammalian cells, this results in a damaging depletion and oxidation of the glutathione pool when ascorbate synthesis is increased by feeding L-GulL (*Bánhegyi et al., 1996*; *Puskás et al., 1998*). Ancestral photosynthetic eukaryotes may have been unable to balance their increasing requirements for ascorbate biosynthesis with maintenance of the redox status within the ER, providing selective pressure to uncouple ascorbate biosynthesis from $H_2O_2$ production.

This hypothesis is consistent with the presence of *GULO* rather than *GLDH* in the glaucophytes. As glaucophytes do not appear to use ascorbate for photoprotection, ascorbate biosynthesis would not have been subjected to the same selective pressures as other photosynthetic eukaryotes. This rationale may also apply to the retention of *GULO* in *Galdieria*, which is likely to have both possessed both *GULO* and *GLDH*. *Galdieria* is photosynthetic and expresses a functional APX (*Sano et al., 2001*), but it is very sensitive to even moderate light intensities and grows primarily in an endolithic environment utilising heterotrophic carbon sources (*Oesterhelt et al., 2007*). Thus, photo-oxidative stress in this environment may be minimal, reducing the selective pressure in *Galdieria* to replace *GULO*.

### Evolution of ascorbate auxotrophy in animals

The evolution of vitamin auxotrophy underpins many important nutritional and ecological interactions between organisms (*Helliwell et al., 2013*). The selective pressures resulting in *GULO* loss in animals represent a combination of the costs of ascorbate synthesis (including detoxification of $H_2O_2$ derived from GULO), the physiological requirements for ascorbate and the ecological factors that determine the supply of dietary ascorbate throughout their life cycle. The development of the photoprotective role of ascorbate in photosynthetic eukaryotes would have significantly altered its availability to many heterotrophic organisms. The leaves of land plants have particularly high cellular concentrations of ascorbate relative to other photosynthetic eukaryotes (*Gest et al., 2013*), which may result from their inability to remove intracellular $H_2O_2$ *via* diffusion to an aquatic medium. Our dataset reveals that in almost all documented cases of ascorbate auxotrophy in animals, the major source of dietary ascorbate derives from GLDH rather than GULO (*Table 1*). This is the case even for insectivorous animals, as insects appear to lack *GULO* and must also obtain ascorbate in their diet, primarily from land plants. Thus, the replacement of *GULO* with *GLDH* in photosynthetic organisms may have ultimately been an important contributory factor in the loss of *GULO* in many animal auxotrophs.

The pseudogenisation of *GULO* in primates, bats and guinea pigs is one of the best known examples of evolutionary gene loss (*Drouin et al., 2011*). Through a wider analysis of ascorbate biosynthesis, we have identified that *GULO* has also been lost in photosynthetic eukaryotes. Photosynthetic eukaryotes functionally replaced *GULO* with an alternative terminal enzyme, *GLDH*, which uncoupled ascorbate biosynthesis from $H_2O_2$ production and potentially aided the important photoprotective role of ascorbate. These developments in photosynthetic eukaryotes may have ultimately contributed to the loss of *GULO* in many herbivorous animals, by influencing their supply of dietary ascorbate.

## Materials and methods

### Bioinformatics

A broad range of eukaryote genomes were selected for detailed analyses (*Supplementary file 5*). Sequence similarity searches were used to identify candidate genes involved in ascorbate biosynthesis. Genomic searches were initially performed using BLASTP with mouse or *Arabidopsis*

**Table 1**. Dietary sources of ascorbate in animal auxotrophs

| Animal auxotroph | Primary dietary source of ascorbate | Ultimate dietary source of ascorbate | Enzyme for ascorbate synthesis | References |
|---|---|---|---|---|
| Primates | Land plants | | GLDH | (*Milton and Jenness, 1987*) |
| Guinea pig | Land plants | | GLDH | |
| Bats | Land plants | | GLDH | (*Birney et al., 1976*; *Milton and Jenness, 1987*; *Cui et al., 2011b*) |
| | Insects | Land plants | GLDH | |
| | Fish | Phytoplankton | GLDH | |
| | Blood | | GULO | |
| Passerine birds | Land plants | | GLDH | (*Drouin et al., 2011*) |
| | Insects | Land plants | GLDH | |
| | Small vertebrates | | GULO | |
| Teleost fish | Zooplankton (crustacea) | Phytoplankton | GLDH | (*Dabrowski, 1990*) |
| | Phytoplankton | | GLDH | |
| Crustacea | Phytoplankton | | GLDH | (*Desjardins et al., 1985*; *Hapette and Poulet, 1990*) |
| Phytophagous insects | Land plants | | GLDH | (*Pierre, 1962*; *Dadd, 1973*) |

Major sources of dietary ascorbate were identified in known animal auxotrophs. This information allows us to assess which terminal enzyme contributed to the production of dietary ascorbate. In nearly all cases the major source of dietary ascorbate is most likely to have been derived from GLDH. Phylogenetic analyses suggest GULO has been lost on multiple independent occasions throughout the Chiroptera (bats). Although ancestral bats may have been primarily insectivores, various sources of dietary ascorbate may have contributed to GULO loss. The passerine birds that are unable to synthesise ascorbate are primarily herbivores or insectivores. However, some members of the *Lanius* genus (shrikes) feed also on small vertebrates, in addition to insects. Most teleost fish are believed to be ascorbate auxotrophs due to loss of GULO. As zooplankton (primarily crustacea) are also ascorbate auxtrophs, phytoplankton are likely to be the ultimate source of dietary ascorbate. Reports suggest the ability of crustacea to synthesise ascorbate is either absent or very weak, although the taxonomic sampling and currently available genomic resources are limited. Most, but not all, phytophagous insects have a dietary requirement for ascorbate, and we did not find GULO in any insect genomes. Note also that some species of insect (e.g., cockroaches) may obtain ascorbate from eukaryote endosymbionts, which may allow them to survive on ascorbate-poor diets.

proteins. All instances of protein absence were confirmed using TBLASTN against the genome with additional searches using sequences from closely related organisms. Sequence similarity searches of transcriptomic datasets were used to identify trends in the presence of ascorbate biosynthesis genes, but were not used to infer absence. Proteins recovered from sequence similarity searches were identified using a combination of BLAST score, manual inspection of conserved residues in multiple sequence alignments and their position in phylogenetic trees generated by both neighbour-joining and maximum likelihood method within the MEGA5 software package (*Tamura et al., 2011*).

For detailed phylogenetic analysis of *GULO* and *GLDH*, multiple sequence alignments were generated using MUSCLE. Poorly aligned regions were removed by manual inspection and the alignments were further refined using GBLOCKS 0.91b to remove ambiguously aligned sites (*Talavera and Castresana, 2007*), resulting in an alignment of 263 amino acids. ProtTest (*Abascal et al., 2005*) was used to determine the best substitution model (WAG with gamma and invariant sites) (*Whelan and Goldman, 2001*). Maximum likelihood phylogenetic trees were generated using PhyML3.0 software with 100 bootstraps. Bayesian posterior probabilities were calculated using BEAST v1.8 (*Drummond et al., 2012*), running for 10,000,000 generations, with a burn-in of 1,000,000 generations. As the phylogenetic relationships within the *GLDH* clade were not well resolved, further unrooted phylogenetic analyses of *GLDH* were performed using an individual multiple sequence alignment to allow more positions to be used.

## Biochemical analyses of ascorbate metabolism in rhodophytes

*Galdieria sulphararia* 074G was grown in Galdieria Medium (GM) (CCCryo, Potsdam-Golm, Germany) at 30°C, light intensity 50 µmol m$^{-2}$ s$^{-1}$. *Porphyra umbilicalis* was collected from Maer Rocks, Exmouth, UK (50° 36′ 31.6′′ N 3° 23′ 27.0′′ W). L-Galactose dehydrogenase activity was measured in ammonium sulphate (50% saturation) precipitates of *Porphyra* thallus protein extracts (*Gatzek et al., 2002*). To

determine the impact of exogenous precursors on ascorbate, *Galdieria* cultures or slices of *Porphyra* thallus were incubated with sugars or aldonic acid lactones (25 mM *Galdieria*, 10 mM *Porphyra*) for 24 hr in GM or artificial sea water (Instant Ocean, Aquarium Systems, Sarrebourg, France). *Galdieria* cultures (15 ml) were harvested by centrifugation and extracted with 0.5 ml 80% methanol containing 0.1% formic acid using sonication in the presence of glass beads. *Porphyra* thallus was powdered in liquid nitrogen followed by homogenisation in 80% methanol (0.1 g thallus in 0.5 ml extractant). Homogenates were centrifuged (10 min at 16,000×*g*, 4°C). Supernatants (100 µl) were dried into glass vials, methoximated trimethylsilyl derivatives were prepared (*Lisec et al., 2006*) and analysed by accurate mass GC-EI-qToF MS (Agilent 7200, Agilent Technologies, Santa Clara, CA, USA). Derivatives were injected (0.2–0.4 µl, 1/2 to 1/50 split ratio) onto a Zebron SemiVolatiles GC column (30 m analytical + 10 m guard length, 0.25 mm internal diameter, 0.25 µm film thickness, Phenomenex, Macclesfield, UK) using He carrier gas (1.2 ml min$^{-1}$). Injector temperature was 250°C and column temperature program was 70°C for 4 min, followed by an increase to 310°C at 15°C/min. The column was held at the final temperature for 6 min. Compounds were fragmented at 70 eV and MS spectra were collected (50–600 amu at 5 spectra s$^{-1}$). Ascorbate is oxidised to dehydroascorbate (DHA) during derivatisation and DHA was identified by co-chromatography and comparison of accurate mass spectra with an ascorbate standard. The position of stable isotope incorporation from [1-$^{13}$C]-D-glucose and [6-$^{13}$C]-D-glucose into ascorbate was identified using mass spectra obtained from injection of [1-$^{13}$C]-ascorbate derivatives. C6 of DHA was found in fragments with m/z values of 157.046 and 245.1029, while m/z 316.1038 contained C1 and C6. $^{13}$C enrichment was assessed by the relative abundance of $m + 1$ for each fragment relative to the $^{12}$C ascorbate standard. $^{13}$C-labelled compounds were obtained from Omicron Biochemicals (South Bend, IN, USA) and all other chemicals were from Sigma–Aldrich (Dorset, UK).

## Molecular analyses

Reverse transcriptase PCR was used to verify the expression of *GULO* in *C. paradoxa* (CCAP 981/1) and confirm its coding sequence. RNA was prepared using the TRIzol method (Invitrogen, Paisley, UK) from *C. paradoxa* cultures grown in standard medium (MWC), at 20°C, 16:8 light:dark, light intensity 50 µmol m$^{-2}$ s$^{-1}$. Reverse-transcriptase PCR was performed using a gene specific primer (GGAACTCCTCGAACTTGGGG) for reverse transcription, followed by amplification of a 1017 base pair region using the following PCR primers: GTCGCCCCTTCTGAGCATAG (forward) and CATGAGCGCGTCGAAGTCT (reverse). NCBI accession number KJ957823.

## *Euglena* transcriptome sequencing

*Euglena gracilis* (strain Z) was grown in Koren-Hutner medium (KH) under continuous illumination (24 µmol m$^{-2}$ s$^{-1}$) at 26°C. RNA was harvested by the TRIzol method and used to prepare cDNA. Paired end reads were generated by Illumina sequencing technology resulting in a total of 193,472,913 reads. De novo assembly was carried out using Trinity (*Haas et al., 2013*), followed by further clustering with TGICL (*Pertea et al., 2003*).

## Alternative routes for ascorbate biosynthesis

The three major routes of ascorbate biosynthesis described in *Figure 1* are well supported by biochemical and molecular evidence. However, there is some evidence to suggest that some classes of eukaryotes may use alternative routes to ascorbate or use multiple routes. These pathways are reviewed comprehensively elsewhere (*Loewus, 1999*; *Smirnoff, 2000*; *Linster et al., 2007*) but the implications for our findings are highlighted below.

### Land plants

Ascorbate in land plants is synthesised predominantly via a non-inversion pathway through GDP-D-mannose and L-galactose in which carbon atom 1 (C1) of the precursor hexose remains as C1 in ascorbate (*Loewus and Jang, 1958*; *Loewus, 1999*). Genetic evidence from a range of ascorbate-deficient *Arabidopsis* mutants indicates that the D-mannose/L-galactose pathway is the primary route of ascorbate biosynthesis (*Conklin et al., 1999*, *2006*; *Dowdle et al., 2007*; *Smirnoff, 2011*). For example, *Arabidopsis vtc2 vtc5* double mutants lacking GDP-L-Gal phosphorylase activity are not viable unless supplemented with ascorbate, suggesting no other route can supply sufficient ascorbate to rescue this defect (*Dowdle et al., 2007*). However, a number of other routes to ascorbate in plants

have been proposed. Plants can potentially use D-galacturonate as an ascorbate precursor, and this alternative route may contribute to ascorbate synthesis in certain tissue types, such as fruits (*Isherwood and Mapson, 1956*; *Badejo et al., 2012*). GDP-L-gulose, formed during the activity of GDP-mannose epimerase in vitro (*Wolucka and Van Montagu, 2003*), could provide L-GulL, which could contribute to ascorbate synthesis via the GULO-like enzymes found in land plants (*Maruta et al., 2010*). Overexpression of *Arabidopsis* GULO-like genes in tobacco cell lines increased cellular ascorbate concentration in the presence of exogenous L-GulL (*Maruta et al., 2010*). Oxidation of *myo*-inositol is another potential source of L-GulL (*Lorence et al., 2004*; *Lisko et al., 2013*). However, more definitive biochemical and genetic evidence for these alternative pathways is required in order to assess whether these pathways contribute significantly to ascorbate biosynthesis in plants. The evidence for the alternative pathways is largely based on increased ascorbate production after ectopic expression of genes or addition of exogenous substrates. This approach leaves open the possibility that these pathways are not active in wild type plants. The critical experiments, in which the proposed enzymes are mutated or knocked out, have not been reported.

Our evolutionary analyses indicate that *GULO* and *GLDH* co-existed in ancestral Archaeplastida, suggesting that multiple pathways of ascorbate biosynthesis were operational in these organisms. Whilst the 'plant pathway' via GLDH is clearly the primary biosynthetic route in extant land plants, the presence of minor alternative pathways such as those listed above may reflect this ancestry of shared pathways.

## Prasinophytes

The genomes of the prasinophyte algae *Ostreococcus* and *Micromonas* are unusual amongst the Viridiplantae in that they lack L-galactose dehydrogenase and GDP-mannose pyrophosphorylase (*Urzica et al., 2012*). However, it is likely that these prasinophytes can generate GDP-D-mannose, either through an unidentified alternative enzyme activity (as proposed in brown algae [*Michel et al., 2010*]) or through the transferase activity of VTC2. Furthermore, we identified L-fucose (6-deoxy-L-galactose) dehydrogenase in all *Ostreococcus* and *Micromonas* genomes. L-fucose dehydrogenase also exhibits activity with L-galactose (*Schachter et al., 1969*) and it therefore may functionally replace L-galactose dehydrogenase in these prasinophytes. Analysis of the Marine Microbial Eukaryote Transcriptome dataset revealed that L-fucose dehydrogenase is present in many other prasinophytes including members of the Prasinococcales, Pycnococcaceae and the Pyramimonadaceae. The only prasinophyte identified with L-galactose dehydrogenase was *Nephroselmis pyriformis* (Nephroselmidophyceae). The Chlorodendrophyceae lineage containing *Tetraselmis* spp. also possess L-galactose dehydrogenase.

## Trypanosomes

GULO from trypanosomes exhibits activity with D-AraL or L-GalL but not L-GulL, and is therefore referred to as L-GalL oxidase (*Wilkinson et al., 2005*; *Logan et al., 2007*; *Kudryashova et al., 2011*). The source of L-GalL in trypanosomes has not been determined. We did not find evidence for L-galactose dehydrogenase or the other enzymes in the D-mannose/L-galactose pathway in trypanosome genomes.

## Fungi

The fungi synthesise a range of ascorbate analogues, including 6-deoxy-L-ascorbate, ascorbate glycosides and the five carbon analogue, D-erythroascorbate (*Loewus, 1999*). Yeasts synthesise D-erythroascorbate from D-arabinose via D-AraL and these final steps are therefore analogous to the D-mannose/L-galactose pathway. Deletion of *ALO1* encoding D-AraL oxidase in *Saccharomyces cerevisiae* results in increased sensitivity to oxidative stress but the mutants are still viable (*Huh et al., 1994*). Until both the biosynthesis and the physiological roles of these analogues are better understood, it is difficult to understand how the biosynthesis of the fungal ascorbate analogues may have evolved.

## Prokaryotes

There is little evidence to suggest that prokaryotes synthesise ascorbate de novo. *GULO* and *GLDH* are essentially absent from prokaryotes. Extensive sequence similarity searches identified a single *GULO* sequence in the cyanobacterium *Rivularia* sp PCC 7116 (WP_015122198.1), but all other cyanobacteria lack *GULO*. An enzyme exhibiting L-GulL dehydrogenase activity has been cloned from *Mycobacterium tuberculosis* and a similar enzyme is present in a range of other prokaryotes (*Wolucka and Communi, 2006*). While the presence of this enzyme suggests some prokaryotes have the capacity for ascorbate synthesis, specific evidence for in vivo ascorbate biosynthesis is lacking. There is a single report indicating that some cyanobacteria contain very low concentrations of ascorbate

(*Tel-Or et al., 1986*), although the analytical method used (2,4-dinitrophenylhydrazine) may not be sufficiently specific for ascorbate at these concentrations (*Roe, 1961*). Prokaryotes lack ascorbate peroxidase and it is likely that early reports of ascorbate peroxidase activity in cyanobacteria were due to the activity of other forms of peroxidase (*Bernroitner et al., 2009*).

## Chromerids

The chromerids are photosynthetic relatives of the apicomplexa (*Janouskovec et al., 2010*) and are the only organisms in our analyses that have acquired a plastid via secondary endosymbiosis and retained *GULO*. *C. velia* utilises the xanthophyll cycle for non-photochemical quenching (*Kotabová et al., 2011*), suggesting that ascorbate performs a photoprotective role in chromerids by acting as a co-factor for violaxanthin de-epoxidase (VDE). We found homologues of VDE and VDE-like proteins in the *C. velia* transcriptome, as well as monodehydroascorbate reductase (MDHAR). The presence of *GULO* rather than *GLDH* in *C. velia* may relate to the unusual mitochondria of apicomplexans. Apicomplexans lack complex I of the mitochondrial electron transport chain (*Danne et al., 2013*), which may have influenced their ability to acquire and/or utilise GLDH, as GLDH is incorporated into complex I in land plants (*Schertl et al., 2012*). However, the genomes of the parasitic apicomplexa all appear to lack both *GULO* and *GLDH*. It should be noted that dinoflagellates also lack mitochondrial complex I but possess *GLDH* (*Supplementary file 2*).

## Acknowledgements

The authors acknowledge funding from the Exeter University Science Strategy Fund, Biotechnology and Biological Sciences Research Council (BB/G021678) and the Natural Environment Research Council (NE/J021008/1). Hannah Florance (Exeter University Mass Spectrometry Facility) provided analytical assistance. The *Euglena* transcriptome was sequenced by the Exeter University Sequencing Service with assistance from Konrad Paszkiewicz and Karen Moore. Access to the Marine Microbial Eukaryote Transcriptome Sequencing Project (MMETSP) was provided by National Center for Genome Resources. This paper is dedicated to the memory of Frank A Loewus (1919–2014) who pioneered research into ascorbate biosynthesis.

# Additional information

## Funding

| Funder | Grant reference | Author |
| --- | --- | --- |
| Biotechnology and Biological Sciences Research Council (BBSRC) | BB/G021678 | Nicholas Smirnoff |
| Natural Environment Research Council (NERC) | NE/J021008/1 | Glen Wheeler |
| University of Exeter | Science Strategy Fund | Nicholas Smirnoff |

The funders had no role in study design, data collection and interpretation, or the decision to submit the work for publication.

## Author contributions

GW, NS, Conception and design, Acquisition of data, Analysis and interpretation of data, Drafting or revising the article; TI, VP, Acquisition of data, Analysis and interpretation of data

# Additional files

## Supplementary files

• Supplementary file 1. Distribution of *GULO* and *GLDH* in opisthokont and apusomonad genomes. Genomes of the opisthokonts (including animals and fungi) were examined for the presence of *GULO* and *GLDH*. Strikethrough indicates non-functional pseudogenes. The absence of *GULO* is well documented in the known ascorbate auxotrophs such as haplorhine primates, guinea pigs, bats, teleost fish and passerine birds. Some fungi (e.g., ascomycetes) use D-arabinonolactone oxidase to produce five carbon ascorbate analogue, erythroascorbate. D-arabinonolactone oxidase has different

substrate specificity to GULO, but exhibits a high degree of sequence similarity and has been classed as GULO in the table.

• Supplementary file 2. Identification of *GULO* and *GLDH* in marine microbial eukaryote transcriptomes. Data from the Marine Microbial Eukaryote Transcriptome Sequencing Project (MMETSP, http://marinemicroeukaryotes.org/) were analysed for the terminal enzymes in ascorbate biosynthesis. This dataset contains 679 transcriptomes from 320 different species. Sequence similarity searches used a stringent length cut off to avoid ambiguous results from incompletely sequenced gene products (minimum length 300 amino acids). Using these criteria, we identified *GULO* or *GLDH* in 165 species. Although the absence of a gene in a transcriptome cannot be used to infer absence, we found no examples of organisms that possess both *GULO* and *GLDH*, even when a more relaxed length criterion was used (minimum length 100 amino acids). Note that the underlined *GLDH* sequences from the ciliates *Myrionecta* and *Strombidinopsis* are 100% identical to sequences recovered from their prey (respectively *Geminigera cryophila* and *Isochrysis galbana*). These sequences may therefore be due to contamination. Alternatively, as both these ciliates exhibit kleptoplasty, the presence of algal *GLDH* sequences in the ciliate transcriptome may also represent examples of plastid-related nuclear genes that are retained and transcribed to aid plastid function (*Johnson et al., 2007*). In the latter scenario, these ciliates could therefore use GLDH to temporarily synthesise ascorbate during plastid acquisition.

• Supplementary file 3. Identification of *VTC2* in marine microbial eukaryote transcriptomes. Data from the Marine Microbial Eukaryote Transcriptome Sequencing Project (MMETSP) was analysed for *VTC2*, encoding GDP-L-galactose phosphorylase, the first committed step in land plant ascorbate biosynthesis. Sequence similarity searches used a stringent cut off to avoid ambiguous results from incompletely sequenced gene products (minimum length 300 amino acids). *VTC2* was identified in 37 species, all of which belong to the Chlorophyta. Genes exhibiting weak similarity to GDP-L-galactose phosphorylase, which may represent homologues of GDP-D-glucose phosphorylase (*Adler et al., 2011*), were not included in these results.

• Supplementary file 4. Distribution of ascorbate biosynthetic genes in Archaeplastida transcriptomes. Rhodophyte transcriptomes from the Marine Microbial Eukaryote Transcriptome Sequencing Project (MMETSP) or Genbank (*Chan et al., 2011*, *2012*) were examined for the presence of ascorbate biosynthesis genes. The rhodophytes transcriptomes all exhibit the pathway found in the genomes of *Cyanidioschyzon merolae*, *Chondrus crispus* and *Porphyridium purpureum*, possessing *GLDH* rather than *GULO*. *C. atmophyticus* is a green alga belonging to the Streptophyte lineage containing land plants and charophyte algae. The *Chlorokybus* transcriptome appears unique amongst the Viridiplantate in that it contains *GULO* rather than *GLDH*. All of the other enzymes of the plant pathway are present.

• Supplementary file 5. Genome resources used in this study. A list of the eukaryote genomes used to study the distribution of genes relating to ascorbate biosynthesis and metabolism.

### Major dataset

The following previously published dataset was used:

| Author(s) | Year | Dataset title | Dataset ID and/or URL | Database, license, and accessibility information |
| --- | --- | --- | --- | --- |
| Keeling PJ | 2014 | Marine Microbial Eukaryote Transcriptome Sequencing Project | http://www.ncbi.nlm.nih.gov/bioproject/?term=PRJNA231566 | Publicly available at the NCBI BioProject database (PRJNA231566). |

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
