## [Decision Letter]

Thank you for sending your work entitled “Evolution of alternative biosynthetic pathways for vitamin C following plastid acquisition in photosynthetic eukaryotes” for consideration at e*Life*. Your article has been favorably evaluated by Detlef Weigel (Senior editor), Joerg Bohlmann (Reviewing editor), and two reviewers. The external reviewers are recognized experts with complementary expertise in the areas of molecular evolution and metabolism.

The Reviewing editor and the reviewers discussed their comments before we reached this decision, and the Reviewing editor has assembled the following comments to help you prepare a revised submission.

All reviewers were very positive about this paper and supported its publication in *eLife*, pending editorial revisions. This is a very strong paper on a very interesting topic, which provides fundamental new insights into the evolution of ascorbate metabolism. The paper presents a comprehensive cross kingdom analysis of the various pathways for ascorbate biosynthesis. It has clearly been diligently researched and is presented in a convincing manner, combining literature knowledge, the vast availability of gene sequences and select biochemical analysis.

1) The authors convincingly address questions 1, 2 and 4 which they set out in the Introduction at a high level. However, question 3 was not addressed adequately. This issue can be solved with editorial revisions, i.e. some rewriting.

2) Similarly, although the authors’ statements concerning ascorbate in *Cyanophora* are actually very careful and accurate, this paragraph should be slightly rephrased for clarity as it may be misunderstood as an overstatement of the results.

In summary, all reviewers agreed that this is a very thorough and careful study which presents very interesting observations concerning the evolution of the biosynthesis responsible for synthesizing one of the most important vitamins in the biosphere. It provides considerable new insight and additionally will likely spur future research aimed at a comprehensive analysis of this important pathway. The paper is very well prepared and presented, and has been a pleasure to read.

---

## [Author Response]

*1) The authors convincingly address questions 1, 2 and 4 which they set out in the Introduction at a high level. However, question 3 was not addressed adequately. This issue can be solved with editorial revisions, i.e. some rewriting*.

The EGT scenario of GLDH acquisition proposed in the manuscript provides a potential explanation for the distribution of the different pathways in photosynthetic eukaryotes. The Archaeplastida synthesise ascorbate via GLDH with no inversion of the carbon chain of D-glucose. EGT of GLDH into photosynthetic eukaryotes with secondary plastids would have enabled them to replace GULO in the animal pathway, resulting in a hybrid biosynthetic pathway in which ascorbate is synthesised via GLDH but with inversion of the carbon chain of D-glucose (i.e. the pathway first identified in *Euglena*). This provides an explanation for the observation that photosynthetic eukaryotes with primary plastids operate a plant-type pathway whilst those with secondary plastids operate euglenid-type pathway. We have amended the Results (in the subsection headed “Distribution of ascorbate-dependent antioxidant systems”) and the Discussion (in the subsection headed “Selective pressures underlying evolution of ascorbate biosynthesis”) to clarify this issue.

*2) Similarly, although the authors’ statements concerning ascorbate in* Cyanophora *are actually very careful and accurate, this paragraph should be slightly rephrased for clarity as it may be misunderstood as an overstatement of the results*.

We have carefully rephrased this paragraph (in the subsection headed “Distribution of ascorbate-dependent antioxidant systems”) to make it clear that our findings relate to the role of ascorbate in *Cyanophora* and that the role of ascorbate in cyanobacteria has been examined by previous researchers.